# Beyond $A^*$: Better Planning with Transformers via Search Dynamics Bootstrapping

**Lucas Lehnert**[*]
University of Saskatchewan
lucas.lehnert@usask.ca

**Sainbayar Sukhbaatar**
FAIR team at Meta

**Andy Su**
FAIR team at Meta

**Qinqing Zheng**
FAIR team at Meta

**Paul McVay**
FAIR team at Meta

**Michael Rabbat**
FAIR team at Meta

**Yuandong Tian**
FAIR team at Meta
yuandong@meta.com

## Abstract

While Transformers have enabled tremendous progress in various application settings, such architectures still struggle with solving planning and sequential decision-making tasks. In this work, we demonstrate how to train Transformers to solve complex planning tasks. This is accomplished by first designing a synthetic language that captures the computation performed by the $A^*$ search algorithm when solving a planning task. Then, an encoder-decoder Transformer model is trained to predict this language, resulting in a language model that can correctly solve novel planning tasks by generating $A^*$'s search dynamics. We fine tune this model to obtain a Searchformer, a Transformer model that optimally solves previously unseen Sokoban puzzles 93.7% of the time, while using up to 26.8% fewer search steps than our $A^*$ reference implementation. Searchformer significantly outperforms baselines that predict the optimal plan directly with a 5–10$\times$ smaller model size and a 10$\times$ smaller training dataset. Lastly, we demonstrate how Searchformer scales to larger and more complex decision making tasks with improved percentage of solved tasks and shortened search dynamics.

## 1 Introduction

Transformer-based architectures (Vaswani et al., 2017) have demonstrated impressive performance in different tasks, including holding conversations at the human level (Shuster et al., 2022; OpenAI, 2022; 2023; Touvron et al., 2023), high-quality image understanding (Caron et al., 2021; Oquab et al., 2024; Assran et al., 2023) and video generation (Singer et al., 2023), multi-modal generation (Girdhar et al., 2023; Radford et al., 2021), and code completion (Roziere et al., 2023; OpenAI, 2021). But despite these successes, Transformer-based architectures and Large Language Models (LLMs) still struggle when responding to tasks that involve multi-step planning (Valmeekam et al., 2023a;b) or higher-order reasoning (Momennejad et al., 2023; Fan et al., 2020).

Various methods have been proposed to move past these limitations. One popular approach is to simulate a thinking process and train a Transformer or language model to generate intermediate "thoughts" before outputting a response. These Chain-of-Thought (CoT) prompting (Wei et al., 2022) and Tree-of-thoughts (ToT) methods (Yao et al., 2023) encourage the model to "think" step by step. While these techniques are often effective, they may lead to worse performance in some cases, for example due to self-enforcing (Huang et al., 2023). Techniques effective on one dataset may not work well on others due to changes in the type of reasoning involved (e.g., spatial reasoning vs. mathematical reasoning). How to enable Transformers and LLMs to plan, solve complex sequential decision-making tasks, and perform reasoning still remains elusive and an active area of research.

---

[*]Lucas Lehnert performed this work as part of the FAIR team at Meta.

**Our work.** We demonstrate how to use Transformers and language modelling to robustly solve complex planning tasks. To accomplish this, we design our own synthetic language that expresses planning—the computation involved in obtaining an optimal plan—in a token sequence. In this synthetic language, we also include task specifications as well as a plan solving the task optimally. Similar to LLMs, we train Transformer neural networks to predict the next token given a sequence of tokens in this synthetic language. With this approach, we demonstrate how to build language models that imitate the computation performed by $A^*$ search (Russell & Norvig, 2021, Chapter 3). Lastly, we present *Searchformer*, a Transformer model that solves complex planning tasks in fewer search steps than our $A^*$ reference implementation. This model is obtained through *search dynamics bootstrapping*, a method that first trains a Transformer to imitate $A^*$'s search process and then fine-tunes the model to find an optimal plan within fewer search steps.

To train a Transformer to perform planning, we consider two dataset variants. The first variant uses token sequences only containing a planning task specification and its optimal solution plan. We refer to this data as *solution-only sequences* and neural networks trained on this data as *solution-only models*. The second variant incorporates an execution trace encoding $A^*$'s search dynamics into token sequence. We refer to this data as *search-augmented sequences* and neural networks trained on this data as *search-augmented models*. Lastly, a pre-trained search-augmented model, that generates $A^*$ execution traces and therefore imitates $A^*$'s computation, is fine-tuned to generate shorter token sequences while still outputting an optimal plan. We refer to this final fine-tuned model as a *Searchformer*. When solving Sokoban puzzles, our Searchformer models solve 93.7% of all test tasks while performing on average 26.8% fewer search steps than our own $A^*$ reference implementation.

Through a sequence of experiments that control task complexity, dataset size, and model size, we demonstrate that including execution traces into the training data increases performance on an independent test task set—despite a $10\times$ to $100\times$ increase in the length of the generated sequences. We find that search-augmented models (that include execution traces into their training data) generate an optimal plan more often on unseen tasks with ten times fewer training sequences than a larger solution-only model (that is trained on sequences only including a task description and task solution). This result highlights the power of including $A^*$'s search dynamics into the training process of Transformer models.

## 2 Related work

While existing works (Trinh et al., 2024; Ruoss et al., 2024) leverage synthetic datasets to learn policies for reasoning, our study is fundamentally different in this regard. We demonstrate how to train Transformers that generate the computation performed by a planning algorithm. The resulting system accepts a planning task as an input prompt and then reasons about this task via generation of an execution trace. Our approach stands in contrast to existing Reinforcement Learning (RL) architectures that focus on autonomous decision-making agents and that use a neural network as a black box system to make predictions. RL algorithms such as AlphaZero (Silver et al., 2018), MuZero (Schrittwieser et al., 2020), and AlphaGeometry (Trinh et al., 2024) optimize a neural network as part of a more complex planning or reasoning algorithm. For example, (Silver et al., 2017) use MCTS (Coulom, 2006) as a policy improvement operator (Sutton & Barto, 2018, Chapter 4) to update the neural network's weights. In contrast, our approach first trains a neural network that imitates the computation performed by a planning algorithm. In this work, the neural network is not part of a more complex planning routine. In a similar vein, the presented search dynamics bootstrapping method is driven by the Transformer model itself and not an improvement operator such as policy improvement. To generalize towards more efficient search patterns, this bootstrapping process uses one of our own pre-trained search-augmented models to generate token sequences that encode novel and more efficient search patterns.

Hierarchical planning methods and temporal abstractions (Sutton et al., 2023; 1999; Dietterich, 2000; Hafner et al., 2022) are methods for abstracting multi-step decision sequences, called skills or options, into subroutines. In contrast, this work does not focus on extracting

skills. Instead, we demonstrate how to generate entire execution traces and optimize their length. Unlike prior work (Yang et al., 2022; Pallagani et al., 2022; Ruoss et al., 2024), the trained Transformer models do not predict just a single action and instead output an entire solution plan for a specific planning task.

The property of generating entire execution traces is perhaps most similar to the Scratchpad method (Nye et al., 2021). In contrast to Nye et al., our study focuses on discovering new and more efficient execution traces instead of solely imitating them. We also do not rely on fine-tuning pre-trained LLMs and instead train every model from scratch on a synthetic language.

Our work bears some similarity with neuro-symbolic systems (Graves et al., 2014; Cai et al., 2017), which build differentiable architectures to mimic the functionality of existing symbolic systems. However, these methods use dedicated components (e.g., explicit memory components, built-in recursion), while Searchformer focuses on next-token prediction only. Here, Searchformer relies on generating long contexts and position embeddings (Chen et al., 2023; Peng et al., 2023) to predict in optimal plan. Ultimately, we aim to shed light on how to build more general language model-based architectures that automatically learn a planning mechanism.

Using Transformer architectures to solve complex sequential decision-making tasks has been studied in prior work in an RL setting (Chen et al., 2021; Janner et al., 2021; Laskin et al., 2023). These works present different methods for modelling trajectories of trial-and-error interactions and focuses on predicting a next action, state, or reward or a combination of them. In contrast, we demonstrate how to use a Transformer to model the search steps involved in computing an optimal plan. MCTSNet (Guez et al., 2018) also attempts to learn the search procedure itself, but still hard-codes the MCTS search procedure into a neural network, which leads to quadratic backpropagation overhead and can only deal with up to 500 step rollouts, while our approach can deal with much longer search execution trace. We demonstrate that Transformers can not only imitate a symbolic planning algorithm but can also be used to discover more efficient heuristics via fine-tuning.

## 3 Problem setup

Figure 1 provides an overview of our synthetic dataset generation process. We consider two domains: maze navigation (Figure 1(a)) and solving Sokoban puzzles (Figure 5 in Appendix B). In maze navigation, the goal is to find the shortest path through an $n$-by-$n$ maze. In Sokoban, a worker can move up, down, left, or right and has to push each box onto a dock to solve the puzzle. An incorrect move may immediately lead to a dead end and thus reasoning across many time steps is required to solve the puzzle. Each state in a puzzle consists of a combination of box and worker positions, making Sokoban computationally more difficult to solve than maze navigation.

### 3.1 Generating execution traces of $A^*$ search.

The $A^*$ algorithm computes an optimal plan by manipulating two sets of nodes: a frontier set containing the current search frontier, and a closed set containing all searched nodes. In the maze example in Figure 1(a), each node corresponds to an empty (non-wall) grid cell. For each node, the algorithm computes a heuristic value and a cost from start value. Which node is searched next is determined by the frontier set, closed node set, heuristic values, and cost from start values (Figure 1(c), left panel). $A^*$'s execution trace is collected by tracking all insertion operations into the frontier and closed set along with heuristic and cost from start values (Figure 1(c), right panel). The right panel in Figure 1(c) illustrates the resulting trace for the maze example shown in Figure 1(b). Each row corresponds either to an insertion of a node into the frontier, indicated by a `create` token, or to moving a node into the closed set, indicated by a `close` token. The resulting plan is then appended to this trace. This trace is constructed such that given any prefix, the state of the $A^*$ algorithm can be reconstructed to resume execution. In maze navigation, $A^*$ uses the Manhattan distance

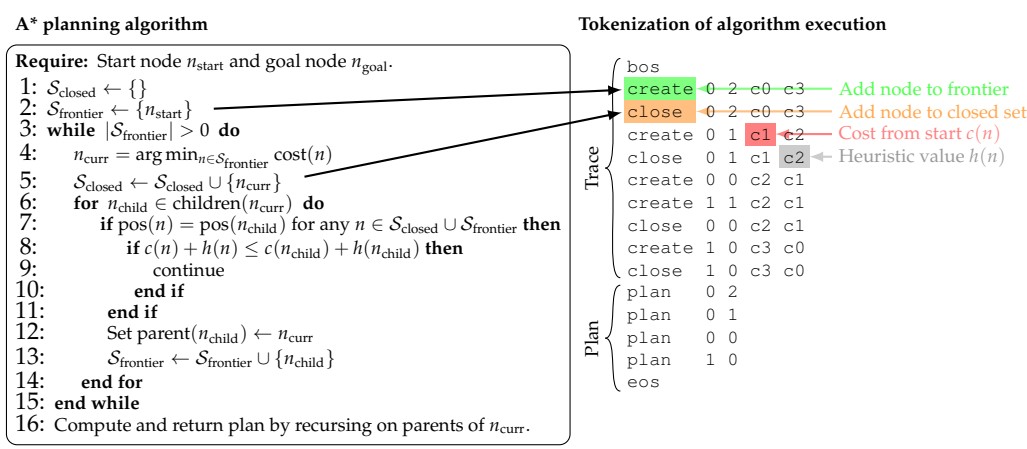

Figure 1: **Expressing a planning task in token sequences.** (a): A $3 \times 3$ maze navigation task where the goal is to find a the shortest path from start to goal without entering a wall cell. (b): The $3 \times 3$ maze navigation task is expressed as a prompt token sequence (left panel) and the optimal plan is expressed as a response token sequence (right panel). The start and end of a sequence is indicated by a beginning-of-sequence token, bos, and an end-of-sequence token, eos. Numbers indicate $x, y$ coordinates. (c): The search dynamics of the $A^*$ algorithm (left panel) is logged into an execution trace (right panel). The first two integers encode the node's $x, y$ coordinates. The last two tokens in the trace encode the cost-since-start value $c(n)$ and the heuristic value $h(n)$ (letter "c" distinguishes costs from coordinate numbers).

to the goal location as a heuristic. In Sokoban, every box is matched to the closest dock and then $A^*$ computes the sum of all Manhattan distances between each box and dock pair.

For each experiment, we generate two token sequence variants, as illustrated in Figure 1:

- *Solution-only sequences* of the format `<prompt><plan>`, where the `<prompt>` part encodes a task description and the `<plan>` part the optimal plan (Figure 1(b)).
- *Search-augmented sequences* of the format `<prompt><trace><plan>`, where the `<trace>` part encodes $A^*$'s execution trace (Figure 1(c)).

Every model is trained from scratch and after training the model's output is parsed and evaluated if it contains an optimal or feasible solution plan.

## 3.2 Training a Transformer model

When generating a token sequence dataset, each task is unique and the test set is constructed such that it does not contain any duplicate of the training set. With this experiment design, we hope to gain insight into how Transformers can be used to solve planning tasks and generalize to previously unseen test tasks.

By including intermediate computation steps into the training data, the Transformer model is trained to imitate the computation performed by the $A^*$ algorithm. In contrast to Procedure Cloning (Yang et al., 2022), the search-augmented model learns to predict the *entire thinking process*, including paths searched by $A^*$ that lead to a dead end.

For each experiment an adaptation of the encoder-decoder T5 architecture (Raffel et al., 2020) is used that integrates Rotary Position Embeddings (RoPE) (Su et al., 2023). The encoder processes the `<prompt>` part of a training sequence, and the decoder processes either a `<trace><plan>`-formatted sequence (search-augmented model) or only a `<plan>`-formatted sequence (solution-only model). Depending on the model variant, each network is trained to maximize the cross-entropy between the distribution of the decoder generations and the distribution of sampling a corresponding sequence from the training dataset. Appendix A describes our optimization setup in more detail.

### 3.3 Moving past algorithm imitation via search dynamics bootstrapping

To reduce the number of tokens generated by a search-augmented model during inference, we implement a method to shift the distribution with which the decoder generates execution traces. First, a search-augmented model is trained to imitate the search dynamics of $A^*$ search. To discover new search dynamics with this search-augmented model and explore the different execution traces, the search-augmented model must generate different sequences for the same task prompt. We accomplish this by inducing non-determinism into the training data and use a non-determinsitic $A^*$ implementation that breaks cost ties randomly and randomizes the order with which child nodes are expanded. This approach does not decrease the efficiency of $A^*$ search itself and merely changes the order with which different nodes are searched while still respecting $A^*$'s heuristic and cost calculations. The resulting search-augmented model will then approximate the probability distribution with which the training sequences were generated.

Once a model is trained to imitate the search dynamics of non-deterministic $A^*$ search, it is used to generate a *new* training dataset consisting of shorter token sequences. This new dataset is constructed by using the trained search-augmented model to sample multiple different token sequences for each training prompt (test prompts are not used to ensure separation between training and test data). Each generated sequence is parsed and checked if it ends in an optimal plan. If this is the case and the sequence is also shorter than the corresponding sequence contained in the original training dataset, then this shortened sequence is included in the new short sequence training dataset. If the generated sequence does not end in an optimal plan or is longer than the original training sequence, then the sequence from the original training dataset is reused.

Subsequently, the search-augmented model is fine-tuned on the new short sequence training dataset. To distinguish from the search-augmented model that imitates $A^*$'s search dynamics, we call this new model *Searchformer*. This procedure can then be repeated by using the resulting fine-tuned model to generate the next even shorter sequence dataset and then fine-tuning the Searchformer model again. In Section 4.3 we demonstrate that this procedure does in fact reduce the number of steps performed during inference while further improving performance. The Searchformer model no longer imitates $A^*$ search and has instead discovered a new way of solving a planning problem using fewer steps.

## 4 Experiments

In our experiments, we use two different $A^*$ implementations for sequence data generation:

1. **Deterministic $A^*$ dataset:** Sequences are generated by executing $A^*$ in a deterministic fashion (by ordering child nodes and breaking equal cost ties deterministically). Given a task prompt, the optimal plan and $A^*$ execution trace is unique, and the Transformer learns the deterministic tie breaking rules implicitly encoded in the data. Evaluating such a model is simple, because the generated sequences need to exactly match the sequences generated by $A^*$.

2. **Non-deterministic $A^*$ dataset:** Sequences are generated by executing $A^*$ in a non-deterministic fashion (by randomly ordering child nodes and breaking equal cost ties randomly). Given a task prompt, the optimal plan and $A^*$ execution trace is no longer unique. Therefore there are multiple correct responses and the Transformer learns to imitate the random tie breaking rules implicitly encoded in the data. The generated

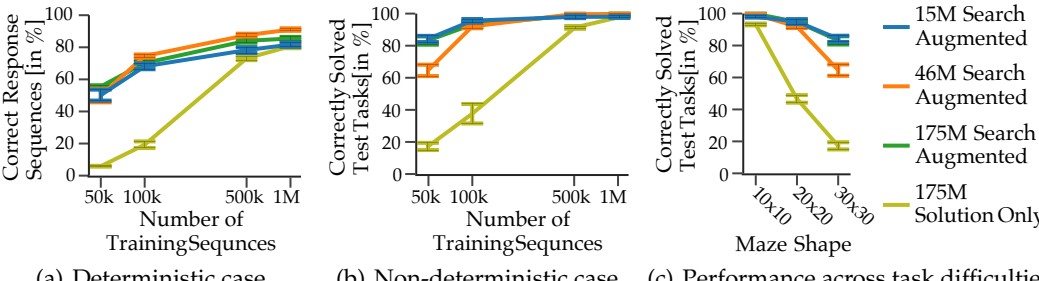

(a) Deterministic case  (b) Non-deterministic case  (c) Performance across task difficulties

Figure 2: **Predicting intermediate computation steps leads to robust performance in the low data regime.** For each model, the number of free parameters (indicated in millions of parameters with "15M", "46M", and "175M") is varied. (a): Comparison of how many test tasks were answered with a correct token sequence when training on the deterministic $A^*$ dataset (exact-match criterion in Appendix C). (b): Comparison for how many test task at least one optimal plan was found when training on the non-deterministic $A^*$ dataset (any-optimal-64 criterion in Appendix C). (c): Performance degradation when increasing task difficulty (maze size). Here, the non-deterministic $A^*$ dataset was used and models are evaluated with the any-optimal-64 criterion.

sequences vary between different executions, but the resulting plans are still optimal and execution traces still respect $A^*$'s cost and heuristic calculations as described in Section 3.3.

Figure 7 (Appendix B) presents an overview of the token sequence length for each dataset and shows that the generated $A^*$ execution traces grow in length with task complexity. Figure 8 (Appendix B) shows that training and test sets are matched in difficulty and have comparable trace lengths. For each task, one model may generate a search sequence ending either in an optimal plan, a feasible plan (a plan that is correct but sub-optimal), or an invalid plan. Appendix C outlines how each model's ability to predict a feasible and optimal plan is scored and how the search dynamics of the search-augmented and Searchformer models are evaluated. Unless indicated otherwise, experiments are repeated five times and figures plot averages across all repeats. All reported errors indicate the Standard Error of Measurement (SEM).

## 4.1 Maze navigation

In the first experiment set, we train a set of encoder-decoder Transformer models to predict optimal plans for maze navigation tasks. We vary the training dataset size and model size (the number of optimized parameters) between different training runs and evaluate each model on the test tasks generated using the same hyper-parameters.

### 4.1.1 Deterministic $A^*$

Figure 2(a) plots for how many test tasks a correct response was generated. Both solution-only and search-augmented models are trained on the deterministic $A^*$ dataset and are evaluated if they exactly re-produce the token sequences generated with $A^*$ search (please refer to the exact-match criterion in Appendix C). Similar to results presented by Ivanitskiy et al. (2023), the solution-only models solve most test tasks for large enough training datasets. Nonetheless, the solution-only models are outperformed by most search-augmented models. Only for large enough training datasets, the solution-only model matches the performance of the worst search-augmented model. In the low training data regime (100,000 training sequences and less), performance of the solution-only model degrades significantly, while the performance of each search-augmented model stays relatively high.

This result is surprising, because for more than 90% of the test mazes, the search-augmented models generate <trace><plan>-formatted sequences that are thousands of tokens long

without predicting any single token incorrectly. Moreover, the solution-only models, that on average predict ten times shorter sequences, are significantly outperformed by the search-augmented models. Even the smallest search-augmented model significantly outperforms the much larger solution-only model parameters. Appendix D.1 presents another ablation experiment testing how sensitive model performance is when the specific tokenization pattern is changed. Here, we find again that the search-augmented models are robust to changes in the specific tokenization pattern and significantly outperform the solution-only.

This result highlights the power of training Transformers to generate long algorithm execution traces. We do not observe compounding prediction errors that usually limit deep model-based RL agents (Asadi et al., 2018), because the used backward-causal decoder network constructs for an $n$-step sequence an $n \times n$ attention map. Here, this property of the Transformer architecture is used to boost performance when predicting an optimal plan.

### 4.1.2 Non-deterministic $A^*$

When trained on non-deterministic $A^*$ data, the model could output multiple different optimal paths for one task. Here, we use each model to generate 64 token sequences for each task. The test task is counted as correctly answered of any one of the 64 sequences contains an optimal plan (please refer to the any-optimal-64 criterion in Appendix C). Because we only test if at least one generated sequence contains an optimal plan, we obtain higher absolute numbers in Figure 2(b) than in Figure 2(a).

Figure 2(b) plots for how many test tasks an optimal plan was found when generating for each test task 64 token sequences. Here, we can observe a pattern similar to the deterministic $A^*$ dataset: even the smallest search-augmented models outperform solution-only model, especially for a small training set. Moreover, we found that model size only impacts the performance of each of the search-augmented models when using very small training datasets (50,000 training sequences). For larger training dataset sizes no significant difference is found. Increasing the number of parameters of the solution-only models does not significantly improve their performance in the low data regime (Figure 9 in Appendix F).

### 4.1.3 Performance across different task difficulty levels and network architectures

Lastly, Figure 2(c) illustrates how a task's difficulty influences the performance of each model. Using the non-deterministic $A^*$ data, we consider the number of correctly solved test tasks as a function of maze size. The larger the maze, the larger the task's state space and the more computation is required to find an optimal solution plan. While the solution-only model's performance drops rapidly as the tasks become more challenging, the search-augmented models maintain a comparably high accuracy, even for the smallest model size. Appendix F presents a full comparison across all maze sizes. Apart from overcoming computational resource constraints, we believe that this approach scales to even more complex tasks.

While the solution-only models learn to predict an optimal plan for large and diverse enough training datasets, search-augmented models perform significantly better in the low data regime for more difficult tasks. The search-augmented models reach higher performance because they perform additional computation during inference. These models imitate the search dynamics for a grounded reasoning chain that leads to an optimal plan. In contrast, the solution-only models have to infer direct correlations between task description and optimal solution plan through supervised training where many of such correlations can be spurious and unreliable during evaluation on the test task set.

To test if the presented results depend on the use of an encoder-decoder architecture, we trained a decoder-only architecture in <prompt><trace><plan>-formatted sequences. This architecture had 46 million parameters and apart from network architecture no other hyperparameters were changed. For this experiment we used 50,000 training sequences from the $20 \times 20$ maze datasets. The resulting decoder-only search augmented model had a test set performance of 94.8% while the corresponding encoder-decoder search augmented models solved 91.8% of the test tasks. This result indicates that our approach can be implemented with different architectures to obtain even better test set performance.

Table 1: **Test set performance in the Sokoban tasks.** Over 200 unseen test Sokoban tasks, we report percentage of solved and optimally solved test tasks. For sequences ending in either an optimal and correct plan we report the SWC (*Success Weighted by Cost*) score, and ILR (*Improved Length Ratio of Search Dynamics*) scores. In every column, higher scores indicate better performance. Please check Appendix C for detailed definitions of these scores.

| Params. | Model | Solved (%) | Optimal (%) | SWC | ILR (solved) | ILR (optimal) |
|---|---|---|---|---|---|---|
| 45M | Solution only | 90.3 ±1.0 | 86.8 ±0.3 | 0.890 ±0.009 | – | – |
| | Search augmented | 92.5 ±1.0 | 90.8 ±1.6 | 0.924 ±0.011 | 0.908 ±0.020 | 0.919 ±0.019 |
| | Searchformer, step 1 | 95.5 ±1.0 | 93.5 ±1.0 | 0.953 ±0.010 | 1.054 ±0.025 | 1.062 ±0.015 |
| | Searchformer, step 2 | 96.0 ±0.5 | 93.4 ±0.6 | 0.957 ±0.005 | 1.158 ±0.025 | 1.181 ±0.012 |
| | Searchformer, step 3 | 95.5 ±0.8 | 93.7 ±1.6 | 0.953 ±0.009 | 1.292 ±0.044 | 1.343 ±0.067 |
| 175M | Solution only | 95.7 ±0.2 | 90.0 ±0.8 | 0.949 ±0.003 | – | – |
| | Search augmented | 95.2 ±0.9 | 93.2 ±1.0 | 0.949 ±0.010 | 0.925 ±0.010 | 0.933 ±0.011 |
| 757M | Solution only | 96.5 ±0.1 | 92.2 ±1.2 | 0.958 ±0.002 | – | – |

## 4.2 Solving Sokoban puzzles

To test if similar results can be obtained on a different and more complex task with a different tokenization pattern and different transition structure, we repeat our experiments for Sokoban puzzles using our non-deterministic $A^*$ implementation. Table 1 lists how often each model generated a correct optimal plan for each test task. As before, by training on execution traces, the search-augmented models outperform the solution-only models. Even increasing the parameterization of a solution-only model to 747 million parameters only leads to a marginal performance improvement. On average, this 747 million parameter solution-only model is still outperformed slightly by a smaller 175 million parameter search-augmented model. This experiment further confirms our findings on more complex planning tasks with a different transition structure and a different tokenization method.

## 4.3 Searchformer: Improving search dynamics via bootstrapping

In this last experiment, we investigate how the search-augmented model can be iteratively improved to compute an optimal plan while generating a shorter execution trace. Here, our goal is to shorten the length of the search trace while still producing an optimal solution.

We start out with the smallest search-augmented model trained on the non-deterministic $A^*$ Sokoban dataset and use it to generate a new shorter sequence training dataset as outlined in Section 3.3. For each Sokoban puzzle in the training data, we generated 32 different `<trace><plan>`-formatted sequences by sampling tokens from the Transformer's output distribution and include the shortest generation (measured in tokens) if it contains an optimal plan. Subsequently, we fine-tune the search-augmented model on this newly created training data (by running an additional 10,000 training steps) to obtain the first Searchformer model. Using this Searchformer model, we subsequently generate another short sequence dataset and repeat the fine-tuning procedure to further improve the model.

Figure 3 illustrates how the sequence lengths generated by the Searchformer model's are iteratively shortened. With every improvement step, the average length of the generated traces—the number of search steps—decreases (Figure 3(a)). The final Searchformer model generates search dynamics sequences that are on average 26.8% shorter than the sequences generated with $A^*$ search. Consequently, the Searchformer model found a way to find a plan in a complex task that is more efficient in terms of search steps than the $A^*$ implementation used to train the initial search-augmented model. The search-augmented models match the response length distribution of our $A^*$ implementation on the test task set, while the Searchformer model's distribution is skewed towards shorter sequence lengths (Figure 3(b)).

As reported in Table 1, fine-tuning resulted in a significant performance improvement, reducing the rate of incorrect and non-optimal solutions by 40% and 30% respectively. The *Success Weighted by Cost* (SWC) score (Wu et al., 2019) factors in how many test tasks were

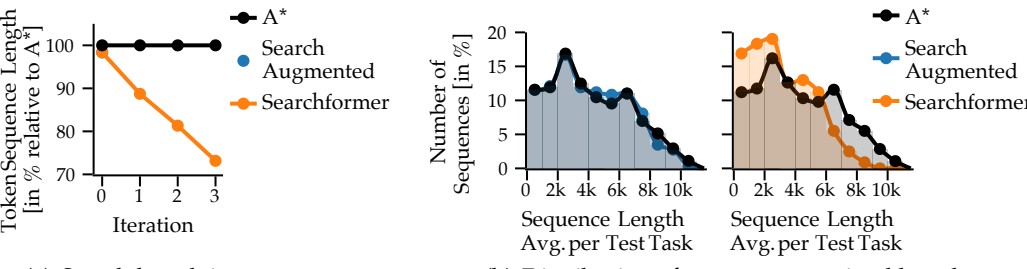

(a) Search length improvement.  (b) Distribution of average-on-optimal length.

Figure 3: **Improvement of search dynamics length via bootstrapping in Sokoban** (a): For each test task answered with an optimal plan, the average generated execution trace length is computed and averaged across experiment repeats. At each iteration, the subset of test tasks answered with an optimal plan by the search-augmented and Searchformer models are compared and the relative improvement over our $A^*$ reference implementation is plotted. Figure 12 provides more details about the sequence length distribution. (b): Distribution of execution trace lengths for each optimally answered test task before fine-tuning (search-augmented model shown in the left panel) and after three fine-tuning iterations (Searchformer shown in the right panel).

solved correctly and how close the predicted plans are to the optimal length (Appendix C). A perfect score would be one, and one can see in Table 1 that the comparably small Searchformer matches the performance of the largest solution-only model (also note the small SEM values). The *Improved Length Ratio of Search Dynamics* (ILR) measures how much the length of each execution trace is shortened (Appendix C). With each improvement iteration the scores increase and climb above one.

The results reported in Figure 3 and in Table 1 compare each model's performance on test tasks that were either correctly or optimally solved. To test if a model overfits to easier test tasks with shorter execution traces, we plot in Figure 10 (in Appendix D) the execution trace length generated with $A^*$ against the execution trace length generated by each model. Each point in this scatter plot corresponds to a single test task. Here, the trend of shortening the execution trace via search dynamics bootstrapping is clear and one can also observe that neither model only specializes on solving easier test tasks with shorter execution traces.

## 5   Discussion

State-of-the-art LLMs exhibit remarkable capabilities in responding to user prompts in a seemingly intelligent manner. Despite these advances, LLMs often provide false or misleading responses when prompted with tasks involving goal-directed reasoning and sequential decision making. For example, state-of-the-art LLMs fail to solve even small maze navigation tasks with in-context learning (Appendix E). To better understand how language models can be used for planning, this work aims to build a connection between language modelling, planning, and sequential decision making. In this context, we demonstrate how to train specialized language models to robustly solve complex planning tasks.

In contrast to classic planning algorithms, formal guarantees about test performance or the number of bootstrapping iterations needed to reach a certain performance level are unknown for our approach. For some small number of test tasks, we found that all models fail to find an optimal or even feasible plan. This property is a consequence of using deep learning techniques and relying on the empirical risk minimization framework (Vapnik, 1991). The aim of this work is to open new doors towards discovering new search heuristics and to develop a better understanding of how language models can be applied to planning problems. The aim is not to replace traditional planning algorithms for which there exist formal guarantees with far more efficient implementations.

### 5.1 Limitations and future work

In this work, a backward-causal decoder is used to generate an entire execution trace. This makes our method computationally very expensive for complex tasks with very long execution traces. In fact, the presented experiments use token sequences that are significantly longer than the sequences used to train LLMs such as Llama 2 (Touvron et al., 2023). One way to mitigate this limitation and improve the efficiency of the presented method is to use curriculum learning: Starting from simple tasks with reasonably long execution traces, train and fine-tune a Searchformer to reduce the trace length, and then use this model to generate shorter execution traces for slightly more complex tasks. Another possibility is to explore other planning algorithms or integrate value functions into $A^*$ search, similar to MCTS, to cap the maximal depth the search algorithm explores. Integrating hierarchical planning methods and temporal abstractions (Sutton et al., 2023; 1999; Dieterich, 2000; Hafner et al., 2022) are another avenue to equip the resulting model with the ability to abstract over decision sequences and find an optimal plan using fewer computation steps.

A key aspect of designing a synthetic language that captures algorithmic computation is ensuring that with any token sequence prefix, the algorithm execution state can be reconstructed and execution can be resumed. Although we focus on $A^*$ search as an example, our approach could be applied to various algorithms and encourages further research in that direction. For example, our data assumes that only optimal plans are encoded. If training data includes sub-optimal but feasible plans, the resulting solution-only and search-augmented models would produce feasible yet sub-optimal plans. This motivates further experimentation with different LLM fine-tuning techniques (Ouyang et al., 2022; Kaufmann et al., 2024) to enhance solution quality.

As illustrated in Figure 10, each bootstrapping iteration improves the Searchformer in terms of the generated execution trace lengths. One direction of future work would be to investigate how many iterations are optimal and to what degree the model can be improved. Here, variance in execution trace length of the initial training data may play an important role. For example, using an inefficient search heuristic will lead $A^*$ to explore larger portions of the search tree until an optimal solution is encountered by random chance, leading to a higher variance in the resulting trace dataset. In this case, improving the Searchformer model is easier. While there might be theoretical limits to how much a particular algorithm can be improved, a Searchformer model may also exploit certain properties of the specific task distribution it is trained on.

In comparison to Plansformer (Pallagani et al., 2022), our approach focuses on training Transformers from scratch to solve complex planning tasks on synthetic datasets. We believe that our results and methods could be combined with methods such as Plansformer to fine-tune LLMs and enable them to solve complex planning tasks more robustly. In this context, one important direction for future work is how our approach transfers to in-context learning in pre-trained LLMs that communicate with human users in natural language.

## 6 Conclusion

We demonstrate how to effectively train Transformers to solve complex planning tasks by designing a synthetic language that mirrors computations performed by the $A^*$ planning algorithm. We also introduce a bootstrapping method to optimize this synthetic language, resulting in a model that solves novel evaluation tasks in fewer search steps than our reference $A^*$ implementation. With this study, we hope to shed light on how language models can be applied to sequential decision making and hope to inform further research about better understanding the reasoning capabilities of LLMs.

### Acknowledgments

We thank Amy Zhang for helpful discussions on earlier versions of this work. We also thank the anonymous reviewers for valuable feedback that strengthened this paper.

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

# A  Training encoder-decoder Transformers to predict optimal plans

All program code used for the experiments presented in this paper as well as all model check-points can be found on Github at `https://github.com/facebookresearch/searchformer`.

In this work, we consider encoder-decoder Transformers (Raffel et al., 2020) and process each prompt—the task specification—with an encoder network to obtain an embedding of a planning task. Using the decoder, the resulting embedding is then decoded into a response of the format `<trace><plan>` or the format `<plan>`. We refer to a model predicting sequences of the format `<trace><plan>` as a *search-augmented* model and a model predicting sequences of the format `<plan>` as solution-only models.

## A.1  Encoder-decoder architecture

A token sequence is processed by a neural network by first indexing the set of all tokens contained in a specific dataset. This indexing is used to map any token sequence to a set of integers. Formally, we denote a prompt as a sequence of integers $x_{1:n} = (x_1, ..., x_n)$. An encoder Transformer neural network with network weights $\boldsymbol{\theta}_{\text{enc}}$ is a function $f_{\boldsymbol{\theta}_{\text{enc}}}$ mapping a tokenized prompt $x_{1:n}$ of arbitrary length $n$ to a sequence of $n$ real-valued vectors:

$$f_{\boldsymbol{\theta}_{\text{enc}}} : x_{1:n} \mapsto \boldsymbol{z}_{1:n} \tag{1}$$

Each vector $\boldsymbol{z}_i$ in this sequence of vectors $\boldsymbol{z}_{1:n}$ has the same dimension. The decoder network is then used to generate a response auto-regressively: Starting with a specific beginning-of-sequence token bos to cue the decoder, a sequence is recursively built by first mapping the start token bos to a probability distribution over next-tokens. This probability distribution is stored as a vector $\boldsymbol{p}_1$ whose dimension is equal to the size of the vocabulary. The next token is then generated by sampling from this distribution and the sampled token is appended to the response sequence. Subsequently the two-token response sequence is fed into the decoder again to compute the next probability vector $\boldsymbol{p}_2$ and sample the next token. This procedure is repeated until an end-of-sequence token eos is sampled. While only the last computed probability vector is needed to sample the next token, the decoder network simultaneously predicts a sequence of next token probability vectors $\boldsymbol{p}_{1:m}$ given an input sequence $y_{1:m}$. Furthermore, this prediction is conditioned on the encoder output $\boldsymbol{z}_{1:n}$. The decoder Transformer neural network with weight parameters $\boldsymbol{\theta}_{\text{dec}}$ is therefore a function

$$g_{\boldsymbol{\theta}_{\text{dec}}} : \boldsymbol{z}_{1:n}, y_{1:m} \mapsto \boldsymbol{p}_{1:m}. \tag{2}$$

The encoder network $f_{\boldsymbol{\theta}_{\text{enc}}}$ and decoder network $g_{\boldsymbol{\theta}_{\text{dec}}}$ internally both use a number of stacked causal attention layers to form an encoder-decoder Transformer (Vaswani et al., 2017) as outlined in more detail in Appendix A.3. We denote a concatenation of all encoder parameters $\boldsymbol{\theta}_{\text{enc}}$ and decoder parameters $\boldsymbol{\theta}_{\text{dec}}$ with the vector $\boldsymbol{\theta}$.

## A.2  Training with teacher forcing

An encoder-decoder architecture is optimized to generate responses that follow the distribution of a training dataset by minimizing the cross-entropy between the training data distribution $p_{\mathcal{D}}$ over prompt-response pairs $(x_n, y_m)$ and the distribution $p_{\boldsymbol{\theta}}$ with which the encoder-decoder model is generating responses. This cross-entropy loss objective

$$H(p_{\mathcal{D}}, p_{\boldsymbol{\theta}}) = \mathbb{E}_{\mathcal{D}} \left[ -\log p_{\boldsymbol{\theta}}(y_{1:m}|x_{1:n}) \right] \tag{3}$$

$$= \mathbb{E}_{\mathcal{D}} \left[ -\sum_{i=1}^{m-1} \log p_{\boldsymbol{\theta}}(y_{i+1:m}|y_{1:i}, x_{1:n}) \right], \tag{4}$$

where line (4) follows from the auto-regressive generation procedure described before. Within the same planning dataset, different prompt-response pairs can have different prompt and response lengths. To emphasize shorter response sequences during training, we re-weigh each sample resulting in the loss objective

$$L(\boldsymbol{\theta}) = \frac{1}{D} \sum_{d=1}^{D} \frac{1}{m_d - 1} \sum_{i=1}^{m_d - 1} \log p_{\boldsymbol{\theta}}(y_{i+1:m_d}^d | y_{1:i}^d, x_{1:n_d}^d), \tag{5}$$

where the first summation averages over all $D$ prompt-response pairs of the training dataset. In Equation (5) the super-script $d$ indexes individual prompt-response pairs in the training dataset. This average is an empirical approximation of the expectation in Equation (3) for a finite i.i.d. sample of size $D$. This loss objective is optimized using gradient descent (Goodfellow et al., 2016, Chapter 10).

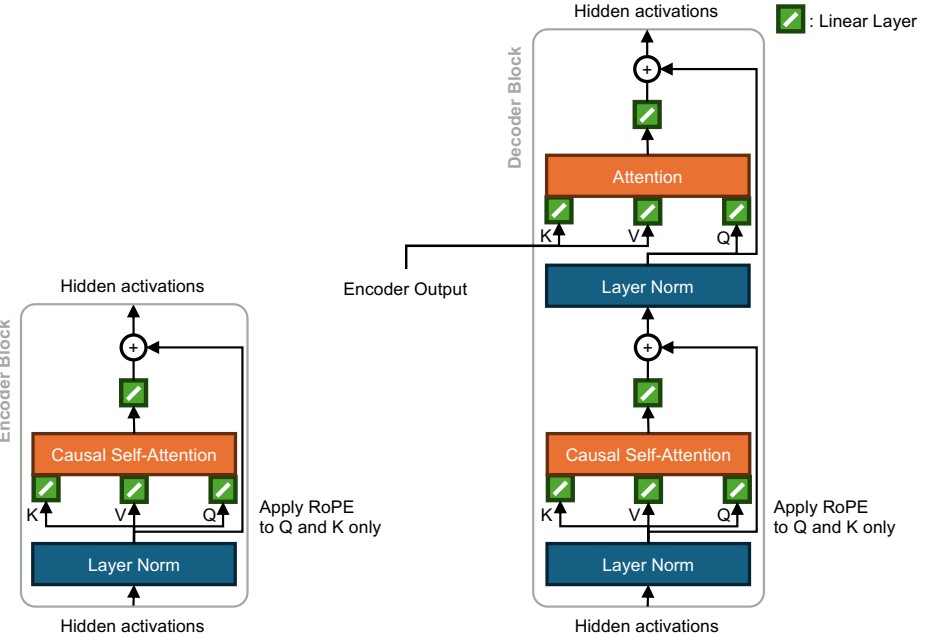

Figure 4: **Attention blocks architecture used in the encoder-decoder Transformer architecture.** The encoder network consists of a set of feed-forward layers where each layer processes hidden activations in parallel with a set of encoder attention blocks (left diagram). Similarly, the decoder network consists of a set of feed-forward layers composed of a number of decoder attention blocks (right diagram). The number of blocks in each layer is referred to as the number of heads. Token sequences are first mapped into integer sequences using a look-up dictionary. Then, these sequences are fed through a PyTorch (Paszke et al., 2019) `torch.nn.Embedding` layer to map the integer sequence into a sequence of hidden activation vectors. After the last decoder layer, hidden activations are mapped through a linear layer into a sequences of logits vectors to compute the next token probability vectors $\boldsymbol{p}_{1:m}$.

### A.3  Network architecture and hyper-parameters

The encoder-decoder Transformer first maps every token to a one-hot vector of the same dimension as the token vocabulary space. These one-hot vectors are then projected through a linear embedding layer into a set of vectors.

The encoder then consists of multiple feed-forward layers and each layer consists of multiple encoder blocks (left part of Figure 4). The output of these layers is then mapped through another linear layer to project the hidden activations into a tensor of the correct shape. The decoder also consists of multiple feed-forward layers and each layer also consists of multiple decoder blocks (right part of Figure 4) processing the hidden activations in parallel. As illustrated in Figure 4, the decoder network is conditioned on the encoder by processing the encoder's output tensor directly in the second attention map. Furthermore, each tokens position is encoded by applying RoPE embeddings (Su et al., 2023) as indicated in Figure 4. We did not use dropout in our architecture.

Table 2 lists the used architecture hyper-parameter and Table 3 lists the hyper-parameters used for optimizing each model. All experiments were implemented in PyTorch 2.0 (Paszke et al., 2019) and default parameters were used unless reported here otherwise.

Table 2: **Architecture Hyper-parameters.** The same parameters were used in both the encoder and decoder network. The number of heads indicates how many attention blocks are used in one layer. Layer dimension indicates the dimension of the feature vectors processed through each attention block (dimension of $K$, $V$, and $Q$ in Figure 4). All models used a RoPE frequency of 10000.

| Parameter | 15M model | 46M model | 175M model | 747M model |
|---|---|---|---|---|
| Layers | 6 | 8 | 9 | 16 |
| Heads | 3 | 4 | 4 | 12 |
| Layer dim. | 64 | 96 | 192 | 96 |

Table 3: **Optimization hyper-parameters.** Every model was optimized using AdamW (Loshchilov & Hutter, 2019) with setting $\beta_0 = 0.9$ and $\beta_1 = 0.99$. Initially, the learning rate was linearly interpolated: Starting at zero and then increasing linearly to the value listed below until step 2000. Then a cosine learning rate schedule was followed (Loshchilov & Hutter, 2016). to the between zero at the first training step and the listed value below for the

| Parameter | Model | Maze Tasks | Sokoban Puzzels |
|---|---|---|---|
| Learning rate | 15M | $2.5 \cdot 10^{-4}$ | $2.5 \cdot 10^{-4}$ |
| | 46M | $7.5 \cdot 10^{-5}$ | $7.5 \cdot 10^{-5}$ |
| | 175M | $5.0 \cdot 10^{-5}$ | $5.0 \cdot 10^{-5}$ |
| | 747M | – | $5.0 \cdot 10^{-5}$ |
| Batch size | all | 16 | 64 |
| Training steps | all | 400000 | 80000 |

## B  Dataset generation

All datasets including token dataset generated through inference with our models are publicly available through our Github repository at `https://github.com/facebookresearch/searchformer`.

All datasets were generated by first randomly sampling a task and then executing $A^*$ to obtain an optimal plan. Maze tasks were generated first by randomly selecting 30-50% of all cells to be wall cells. Then a start and goal location was randomly selected and $A^*$ was executed to obtain an optimal plan. If the plan had a length of at least the mazes width or height (e.g. for $10 \times 10$ mazes the optimal plan needs to contain at least 10 steps), then the task was added into the dataset. For Sokoban, a $7 \times 7$ grid was sampled and two additional wall cells were added as obstacles to the interior of the map. Then two docks, boxes, and the worker locations were randomly placed. If the sampled task is solvable by A*, then the task was admitted to the dataset.

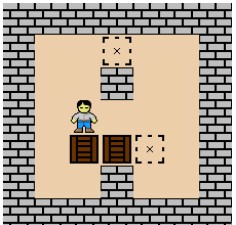

Figure 5: **Example Sokoban puzzle[2].** The prompt, $A^*$ execution trace, and optimal plan for this task is illustrated in Figure 6 in Appendix B. For Sokoban, we only use a non-deterministic $A^*$ implementation.

Due to the large volume of generated data, all data was stored in and transformed with a MongoDB (MongoDB Inc.) instance and map-reduce techniques. Furthermore, when sampling tasks, the dataset was constructed to reject duplicate tasks ensuring that training and test data and each prompt is distinct. Once each task and trace dataset was generated, each execution trace is converted into prompt and response token sequences, as illustrated

---

[2]This level image was composed using image icons from `https://github.com/morenod/sokoban` (accessed 2023-11-21).

in Figure 1(c) and Figure 8. Because the Sokoban tasks are very complex and difficult to solve for A*, the resulting token sequences were partially very long and reached almost 100000 tokens. Due to GPU memory requirements, the Sokoban dataset was further sliced to only include sequences of with at most 10000 tokens. Figure 8 compares the sequence length distribution for each dataset. During training, each dataset was also sorted and sliced to only contains the reported number of training sequences. Furthermore, each model was evaluated on the same test tasks within each task dataset. The test dataset contains plans and traces that are of comparable length to the training data (Figure 8).

```
Prompt {
  bos                                    Trace      bos
  worker 2 3                          (2583 tokens) create worker 2 3 box 2 4 box 3 4 c0  c3       plan 2 3
  box    2 4                                        close  worker 2 3 box 2 4 box 3 4 c0  c3       plan 1 3
  box    3 4                                        ...                                            plan 1 4
  dock   1 3                                        create worker 5 4 box 2 3          c10 c3      plan 1 5
  dock   4 4                                        close  worker 2 1                  c12 c0      plan 2 5
  wall   0 0                                                                                Plan { plan 2 4
  wall   0 1                                                                                       plan 3 4
  ...                                                                                              plan 2 4
  wall   6 6                                                                                       plan 2 3
  eos                                                                                              plan 2 2
                                                                                                   plan 1 2
                                                                                                   plan 1 1
                                                                                                   plan 2 1
                                                                                                   eos
}
```

Figure 6: **Token sequence example for Sokoban** This figure lists the token sequence for the Sokoban level depicted in Figure 5.

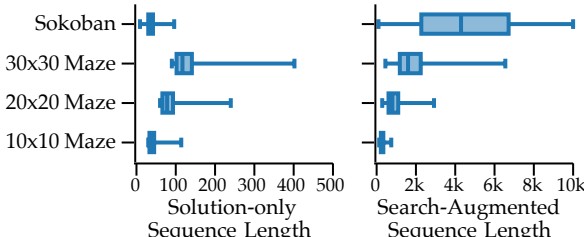

Figure 7: **Training Sequence Length Comparison.** The left panel plots the length of the solution-only sequences and the right panel plots the length of the search-augmented sequences, excluding the start and end of sequence tokens bos and eos. The whiskers indicate the range of all sequence lengths and the box plots indicate the 25%, 50%, and 75% quantiles. Because we focus on planning in complex sequential decision making tasks, the token sequences are multiple orders of magnitude longer than usual token sequences used to train LLMs—especially when $A^*$ execution traces are included in the responses. For example, fine-tuning of the Llama 2 model on human preference data is performed with sequences that are on average 600 tokens long (Touvron et al., 2023).

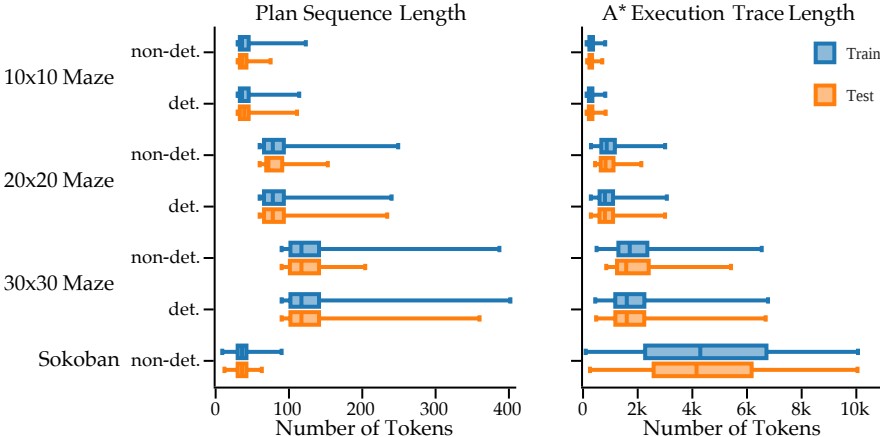

Figure 8: **Sequence length distribution for each dataset.** The training and test sets are designed such that their sequence lengths match closely.

## C   Evaluation criteria

In the presented experiments we evaluate if each model outputs an optimal or feasible plan and how long the generated search sequences are for the search-augmented and Searchformer models.

### C.1   Measuring plan optimality

We evaluate whether the output plan is optimal using one of three criteria:

- **Exact-match criterion.** For each task, if the generated sequence from a trained model matches the output of deterministic $A^*$ exactly, it is labelled as correct, otherwise labelled as incorrect. This is only used to evaluate supervised cloning of deterministic $A^*$.
- **Any-optimal-64 criterion.** For each task, we sample 64 responses from a trained model. Each response is parsed and evaluated if it contains a feasible or optimal plan, regardless of the generated <trace> part. If any of the 64 plans is feasible and optimal, then the task is labelled as correct.
- **SWC score.** To further measure the sub-optimalness of the resulting plans, we also report the *Success Weighted by Cost (SWC)* ([Wu et al., 2019](#)), a statistic that factors in how close the cost $l_i$ of the best predicted correct plan (over 64 trials) is to the optimal plan cost $l_i^*$, averaged across all $n$ test tasks and weighted by a binary variable $c_i \in \{0, 1\}$:

$$\text{SWC} := \frac{1}{n} \sum_{i=1}^{n} c_i \frac{l_i^*}{\max\{l_i, l_i^*\}}.$$

When computing the SWC score, the binary variable $c_i$ is set to one if a correct plan was found and zero otherwise. This SWC score lies between zero and one. If all generated sequences end in an optimal plan, then this value is one.

### C.2   Measuring search dynamics length

For sequences ending in an optimal or feasible plan, we evaluate the length of sequence dynamics in terms of number of tokens using one of the two criteria:

- **Average-on-optimal length.** For each task, we sample 64 responses from a trained model and compute averaged search dynamics length for sequences that lead to an optimal plan.
- **ILR score.** To further measure how much improvement of the model-generated search dynamics against $A^*$ planning, we report the *Improved Length Ratio of Search Dynamics* (ILR) score. Specifically, for each test task $i$, we compute the ratio between the length $t_i$

of the shortest generated search dynamics sequence and the $A^*$ token sequence length $t_i^*$. We then average across all test tasks while only including ratios for tasks for which either an optimal or a correct (and potentially sub-optimal) plan was found, as specified by the binary variable $c_i$. The corresponding measures are thus called *ILR-on-optimal* and *ILR-on-solved*. The ILR is defined as

$$\text{ILR} := \frac{1}{n} \sum_{i=1}^{n} c_i \frac{t_i^*}{t_i}.$$

The ILR measure can take non-negative values and values above one indicate that the model generates shorter search dynamics than the $A^*$ reference. Consequently, if the numbers lie significantly above one, then the model has found a more efficient way to search a task's state space to compute an optimal plan.

## D    Searchformer performance analysis

In Section 3.3, each search-augmented and Searchformer model is evaluated by generating 64 token sequences for each Sokoban test task. For the same test task the same model can generate sequences that end in an optimal plan, an incorrect plan, or a correct but sub-optimal plan. In Figure 10 we compare the length of the generated sequences with the length of sequences generated when using $A^*$ search for each case. The percentages in each panel's caption list how many out of all generated sequences end in an optimal plan, a correct plan (that can be optimal or sub-optimal), and a correct but sub-optimal plan.

In the left panel of Figure 10(a) and Figure 10(b), points are centered around the diagonal axis indicating that the search-augmented models do approximately match the $A^*$ search algorithm in terms of token sequence lengths. Figure 10(a) and Figure 10(b) further confirm the results presented in Sec. 3.3: With every improvement step, the points move down and below the diagonal line. This highlights that the improved Searchformer models generate token sequences that are shorter than sequences generated with $A^*$ search. The Searchformer has found a method of searching a task to compute an optimal plan in fewer search steps than $A^*$ search uses.

Figure 10(c) illustrates what happens when each model generates a correct but sub-optimal plan. Here, the search-augmented model, that is trained to imitate $A^*$, generates trace sequences that are significantly longer than the sequences generated with $A^*$. This suggests that the model struggles in computing a correct plan and generates too many search steps, ultimately leading in finding a correct but sub-optimal plan. Similarly, the Searchformer models also generate sequences that are longer than the sequences generated with $A^*$ search. Despite these inefficiencies, our bootstrapping method is still able to improve the model and bring the average sequence length closer to the length of sequences generated with $A^*$ search (right most panel in Figure 10(c)). While we would desire the trace length to be low in either case, we found that each model generates a correct but sub-optimal plan with less than 5% chance. Consequently, Figure 10(b) shows that the final Searchformer model still generates a correct plan with on average fewer search steps than $A^*$ search. Statistically, the differences between Figure 10(a) and Figure 10(b) are marginal.

### D.1    Ablation experiment comparing plan tokenization pattern

This ablation experiment test how sensitive our performance results are to changes in how plans are encoded in token sequences. For this experiment we use the deterministic A* dataset for 20 mazes and encode a solution plan as a decision sequence instead of encoding the plan as a position sequence. For example, a token sequence of the form `plan 0 2 plan 0 1 plan 0 0 plan 1 0` to a token sequence of the form `plan down down left`. We refer to this tokenization pattern as the *action token style* and emphasize that the tokens `plan`, `up`, `down`, `left`, and `right` are only used to encode the optimal plans. The do not occur anywhere else in the used training sequences.

Table 4 compares the test set performance of the solution-only and search-augmented models for each tokenization style. One can observe that the solution-only models, which

Table 4: **Comparison between different plan tokenization styles** This table compares how changing the plan tokenization pattern impacts performance. This experiment uses the deterministic A* dataset for $20 \times 20$ mazes and compares performance numbers for a 100,000 training sequence dataset. Each number lists the number of correct response sequences (in %) for the test task set. This table reports averages and SEMs across five repeats.

| Params. | Model | Position token style | Action token style |
|---|---|---|---|
| 15M | Solution only | 59.9 ±4.2 | 31.8 ±5.1 |
| | Search augmented | 83.0 ±2.0 | 84.2 ±1.3 |
| 45M | Solution only | 67.7 ±4.2 | 48.2 ±3.8 |
| | Search augmented | 88.1 ±1.8 | 90.8 ±0.7 |
| 175M | Solution only | 68.6 ±4.1 | 38.1 ±7.0 |
| | Search augmented | 88.2 ±1.7 | 93.0 ±0.3 |

are trained on `<prompt><plan>`-formatted sequences, degrade in performance significantly. In comparison, the performance of the search-augmented models, which are trained on `<prompt><trace><plan>`-formatted sequences, increases by a few percent points. This result underlines the benefits of our approach of including execution traces into the training data, leading to datasets with more tightly correlated token sequences that are easier to learn.

## E State-of-the-art LLMs on Maze Task

Below we list examples that even state-of-the-art LLMs still fail to solve a simple maze navigation problem.

**Zero-Shot-Example:**

**Prompt:**
Here's the generated 10x10 maze along with Locations of Walls: (0, 6) (0, 8) (1, 6) (3, 0) (3, 1) (3, 9) (4, 6) (4, 8) (4, 9) (5, 7) (6, 7) (8, 2) (8, 8) (9, 2) (9, 5) (9, 6) (9, 7) (9, 8). Start Location: (0, 0). End Location: (9, 9). Could you solve this? Give me the optimal path in (x,y)

**Gemini Answer:**
$(0, 0) \rightarrow (1, 0) \rightarrow (2, 0) \rightarrow (3, 0) \rightarrow (3, 1) \rightarrow (3, 2) \rightarrow (4, 2) \rightarrow (5, 2) \rightarrow (5, 3) \rightarrow (5, 4) \rightarrow (6, 4) \rightarrow (6, 5) \rightarrow (7, 5) \rightarrow (7, 6) \rightarrow (8, 6) \rightarrow (8, 7) \rightarrow (8, 8) \rightarrow (9, 8) \rightarrow (9, 9)$

**Note:** The path incorrectly includes (3, 0) and (3, 1), (8, 8), (9, 8) which are wall locations.

**Few-Shot-Example:**

**Prompt:**
Here's the generated 10x10 maze along with Locations of Walls: (0, 6) (0, 8) (1, 6) (3, 0) (3, 1) (3, 9) (4, 6) (4, 8) (4, 9) (5, 7) (6, 7) (8, 2) (8, 8) (9, 2) (9, 5) (9, 6) (9, 7) (9, 8). Start Location: (0, 0). End Location: (9, 9). Could you solve this? Give me the optimal path in (x,y)

*Example 1:*
Locations of Walls: (0, 4), (0, 5), (0, 8) (1, 0), (1, 3), (1, 6) (3, 1) (4, 0), (4, 1), (4, 3), (4, 5), (4, 9) (5, 6), (5, 8) (6, 1), (6, 2), (6, 4), (6, 9) (7, 9) (8, 4), (8, 5), (8, 9) (9, 1). Start Location: (0, 0). End Location: (9, 9). Optimal Path: (0, 0), (0, 1), (0, 2), (1, 2), (2, 2), (2, 3), (2, 4), (2, 5), (2, 6), (2, 7), (3, 7), (4, 7), (5, 7), (6, 7), (6, 8), (7, 8), (8, 8), (9, 8), (9, 9)

*Example 2:*
Locations of Walls: (0, 3) (0, 6) (1, 1) (1, 4) (1,7) (3, 4) (4, 2) (4, 6) (5, 0) (5, 4) (5, 6) (6, 6) (6, 7) (6, 8) (6, 9) (7, 3) (7, 6) (8, 0) (8, 4) (9, 1). Start Location: (0, 0). End Location: (9, 9). Optimal Path: (0, 0) (0, 1) (0, 2) (1, 2) (1, 3) (2, 3) (2, 4) (2, 5) (3, 5) (4, 5) (5, 5) (6, 5) (7, 5) (8, 5) (8, 6) (8, 7) (8, 8) (8, 9) (9, 9)

**Gemini Response:**
(0, 0), (1, 0), (2, 0), (2, 1), (2, 2), (3, 2), (3, 3), (3, 4), (3, 5), (3, 6), (4, 6), (5, 6), (5, 5), (6, 5), (6, 4), (7, 4), (7, 5), (7, 6), (7, 7), (8, 7), (8, 6), (8, 5), (8, 4), (8, 3), (8, 2), (8, 1), (8, 0), (9, 0), (9, 1), (9, 2), (9, 3), (9, 4), (9, 5), (9, 6), (9, 7), (9, 8), (9, 9)

**Note:** The path breaks into the walls (4,6) (8,2) (9,2) (9,6) (9,7), (9,8)

## F   Supplemental figures for experiments

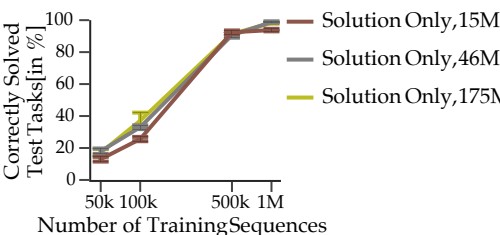

Figure 9:  **Solution-only model performance.** Performance of the solution-only models is primarily influenced by the number of training sequences. Increasing a model's size does not always improve performance.

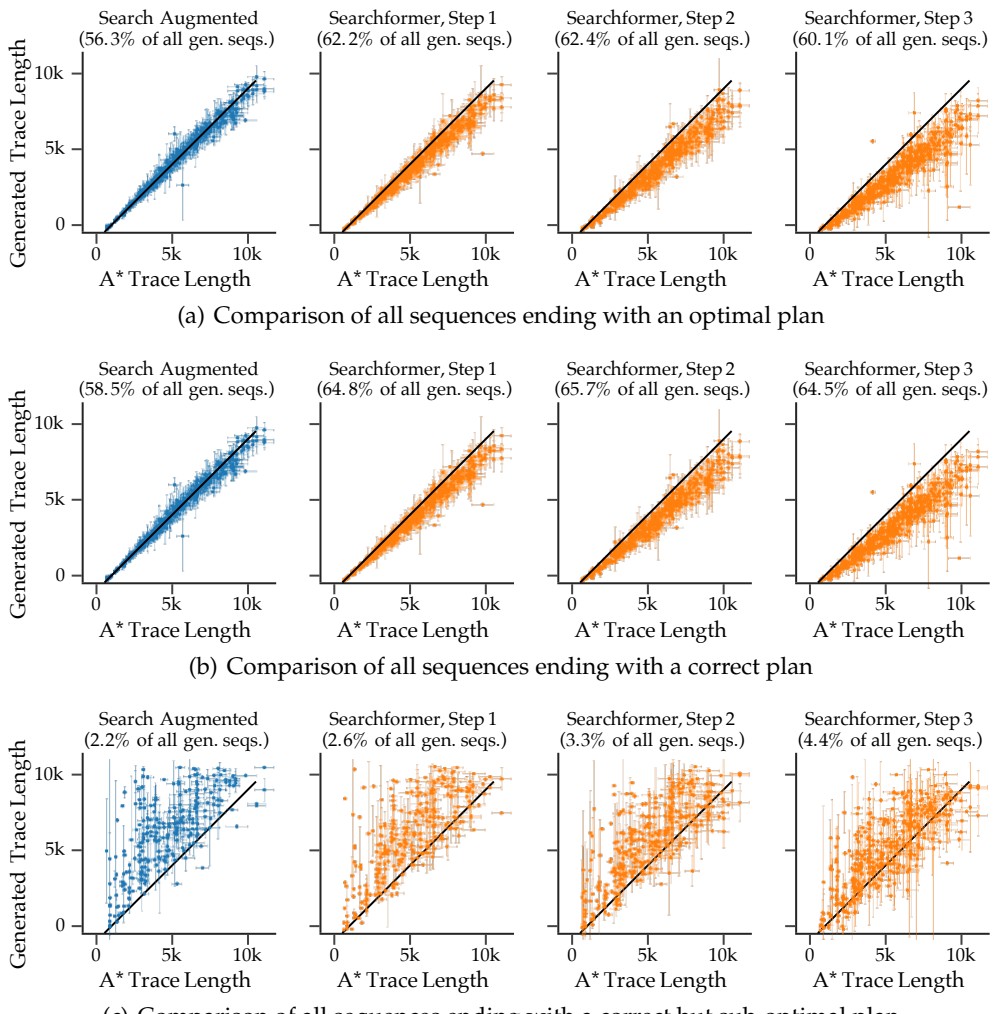

Figure 10: **Sequence length comparison between sequences generated with $A^*$ search and sequences generated with each model.** Each dot in each scatter plot corresponds to one specific test task. On the $x$-axis, we plot the average token sequence length when $A^*$ search is used. On the $y$-axis, we plot the average token sequence length when each model is used. Error bars indicate standard deviations. Percentages indicate the fraction of the generated sequences that are included in each scatter plot. (a): Sequence length comparison for all test prompts for which an optimal plan was generated. (b): Sequence length comparison for all test prompts for which a correct plan was generated. This plot aggregates across sequences ending in an optimal plan and sequences ending in a correct but sub-optimal plan. (c): Sequence length comparison for all test prompts for which a correct but sub-optimal plan was generated using the corresponding model. This plot only aggregates across sequences ending in a correct but sub-optimal plan.

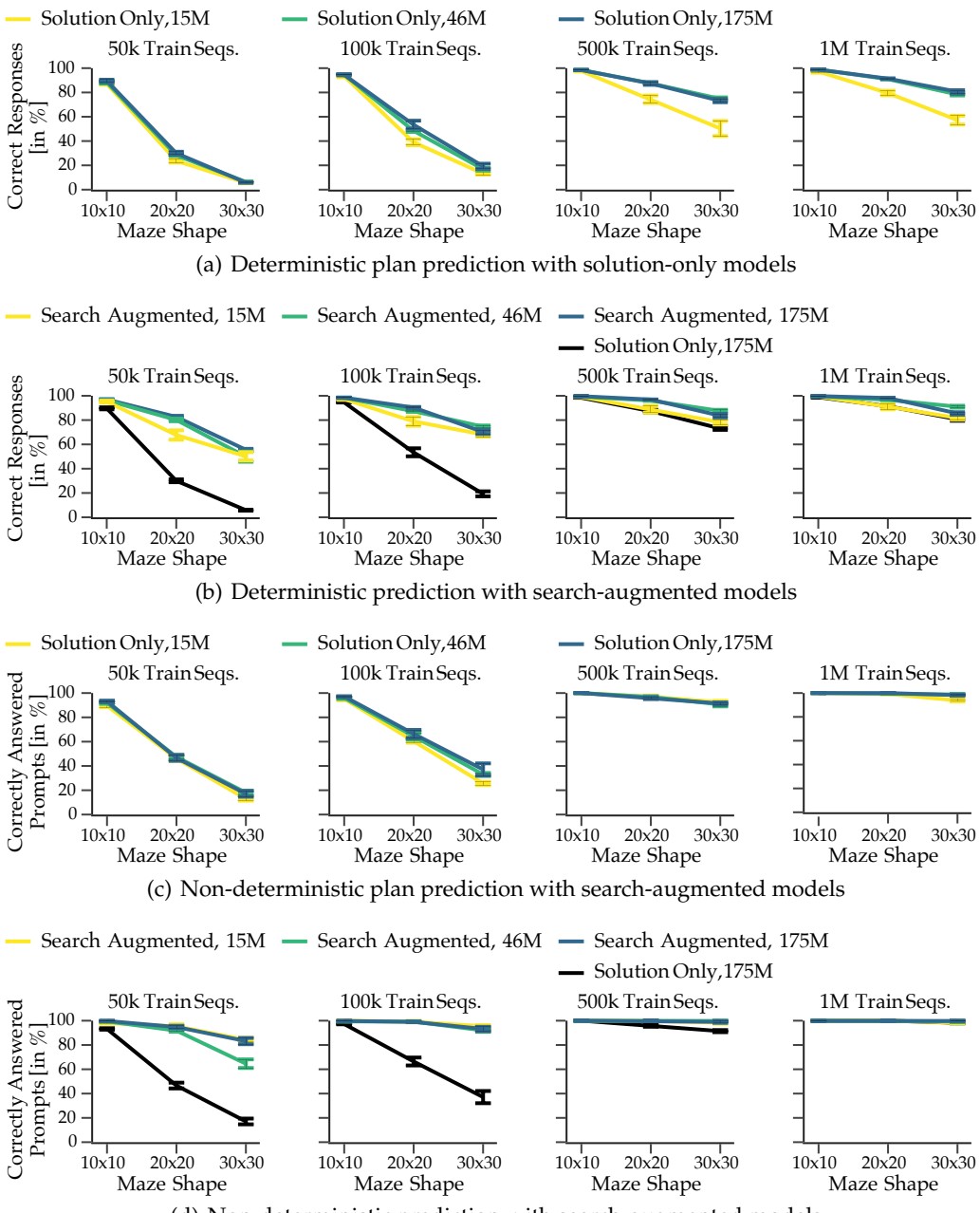

Figure 11: **Optimal plan prediction performance for solution-only and search-augmented models on maze datasets.** (a),(b): Percentage of correctly generated responses. Note that the solution-only models only need to generate a few hundred token long plan to answer a prompt correctly. The search-augmented model must generate a much longer A* execution trace correctly to correctly answer a prompt. This requires the trace plan models to generate response sequences that are hundreds or thousands of tokens long (*cf.* Figure 7). If one token is incorrectly predicted, the response is counted as incorrect. (c),(d): Percentage of correctly generated plans. Each model was used to generate multiple responses for each prompt. If one of the generated responses contains an optimal plan, the test prompt is counted as correctly answered.

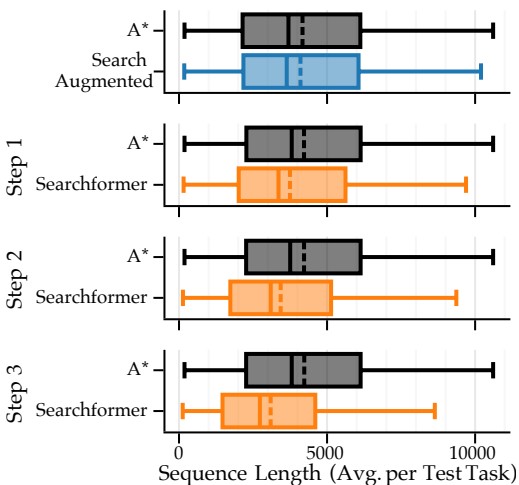

Figure 12: Comparison of search dynamics length (in terms of number of tokens) between $A^*$ search and our models (search-augmented in blue, Searchformer step 1-3 in orange), on the test subset in which our models yield optimal plans. Here, for each test task, we average the lengths of sequences that ends in an optimal plan, out of 64 trials (i.e., average-on-optimal length (Appendix C)). The box plots show the distribution of average-on-optimal lengths, in which the left boundary, mid solid line, right boundary represents the 25%, 50% and 75% quantiles. Dotted lines are means and whiskers show the range.

