# OpenReview forum: "Beyond A*: Better Planning with Transformers via Search Dynamics Bootstrapping"
_colmweb.org/COLM/2024/Conference — COLM_

### Official Review · Reviewer_pZre · 2024-05-10

**Rating:** 9
**Confidence:** 4
**Ethics Flag:** 1

**Summary:**

This paper presents Searchformer, a fine-tuned T5 model for planning. The key findings are that (1) training a transformer to predict not only the plan actions but also the A* search dynamics (called "search-augmented sequence"), offers better (data and model size) efficiency in training and results in more generalizable planners; and that (2) further fine-tuning the transformer on shorter plans that are non-deterministically synthesized yields faster planners.

**Reasons To Accept:**

The paper presents many interesting findings. The idea of training a transformer to predict the search dynamics is novel yet intuitive. It is also surprising to see that a transformer can precisely predict the dynamics consisting of thousands of tokens. The non-deterministic experiment is also interesting; it shows the potential for transformers to discover better plan solutions themselves.

**Reasons To Reject:**

I don't see obvious reasons to reject the paper, but a few thoughts are:
- Generalizing the idea to more complicated navigation tasks. For example, it can be helpful to analyze Searchformer on larger maze environments with obstacles following geometric shapes, which could potentially make the desired trajectories longer. In such complicated environments, does Searchformer still work?
- How sensitive is Searchformer to the specific output representation in Figure 1? For example, instead of using (x, y) coordinates, the plan can also be described as "down, down, right", which is even shorter than the sequence of coordinate tokens. Did the authors see any impact from following different representations?
- Does the idea of Searchformer apply to transformer architectures beyond T5 (e.g., decoder-only LMs)? And does the same idea apply to in-context learning of LMs?

---

> ### Author Rebuttal · Authors · 2024-05-31
>
> We thank the reviewer for the positive review and insightful comments.
>
> **Re. more complex tasks:** In Fig 12, we already present a study where maze size and complexity is scaled, resulting in longer execution traces (Figs 7 and 8). As tasks become more complex, more training data is needed to reach a certain performance level. Apart from computational resource constraints, our results do not indicate that there is any upper limit on task complexity. Searchformer should still work in more complex environments. We will highlight this point in Sec 4.1.3.
>
> **Re. specific output representations:** The exact tokenization scheme does influence the model’s performance. We ran an experiment with the deterministic A* dataset for 20x20 mazes and changed the plan text representation as suggested in the comment (e.g. "plan down down left"). We refer to this as "action token style".
>
> With the "action token style", the 45M search-augmented model's performance increases slightly (from 88.1%±1.8% to 90.8%±0.7%), while the solution-only model's performance drops significantly (from 67.7%±4.2% down to 48.2%±3.8%). This shows that Searchformer is insensitive to the exact tokenization scheme, due to incorporated execution traces that lead to more tightly correlated tokens sequence that are easier to learn, while search-only models may be very sensitive to it. Furthermore, 15M and 175M models show similar trends. We will put full results in the Appendix.
>
> **Re. decoder-only architectures:** We trained one 46M decoder-only search-augmented model on <prompt><trace><plan>-formatted sequences using our 20x20 maze dataset (50k training sequences generated with non-det. A*). This model solved 94.8% of all test tasks with an optimal plan, while the corresponding 45M encoder-decoder search-augmented model solved 91.8% of the test set. This result suggests that our approach also works with a decoder-only architecture. We will include this result in the discussion of our experiments.
>
> **Re. in-context learning:** While the focus of the submitted paper is on training LMs from scratch on synthetic data, we believe that understanding how our approach transfers to in-context learning in pretrained LLMs is an important direction of future research to improve the planning capabilities of LLMs. Our tentative results show SoTA LLMs fail to solve our tasks with in-context learning, even using our text representation. We will revise the future work section to incorporate this point.

---

> > ### Comment · Reviewer_pZre · 2024-06-05
> >
> > Thank you for your response and for adding the new experiments! The results look very interesting; please include them in your next version. I keep my positive rating on this work.

---

### Official Review · Reviewer_6qVT · 2024-05-10

**Rating:** 7
**Confidence:** 4
**Ethics Flag:** 1

**Summary:**

This paper presents SearchFormer, a Transformer architecture trained to solve multistep planning and reasoning tasks. SearchFormer is retrained on token sequences representing the dynamics of A* search and then fine-tuned via expert feedback. SearchFormer successfully solves Sokoban puzzles in fewer steps than standard A*.

This is a novel and interesting use of Transformers, and overall I found the results to be convincing. Some shortcomings that I see concern the lack of comparison to other pathfinding algorithms besides A*, and a lack of guarantees similar to those provided by A*. The significance of the work is undercut by these absences, because without them it comes across as only slightly more than a toy demonstration of Transformers' ability to generate A* sequences and thereby solve A* problems, similar to ChessGPT's ability to generate chess moves and thereby passably play chess.

The paper is also written quite clearly and is easy to follow.

**Questions To Authors:**

A* can guarantee a shortest path, and this property is presumably implicitly captured in the training data. Can SearchFormer similarly guarantee a shortest path, in fewer steps than A*?

How does SearchFormer compare to other pathfinding algorithms? Presumably if it outperforms A* it would also outperform Dijkstra's algorithm, but what about other algorithms like D* where constraints vary?

One question regards relevance to the Conference on Language Modeling: not all Transformers are language models; they just happen to have arisen from NLP and are adept at modeling sequences. Does this research tell us anything new about language or language models specifically?

**Reasons To Accept:**

I find this to be an interesting application of Transformers. Much of the success of Transformers in non-NLP domains has been in representing in the input in as much of a "language-like" fashion as possible vis-à-vis token sequences. This provides more evidence in that direction. If there is a "language" in pathfinding sequences generated by algorithms like A*, then it appears to be discernible by Transformers.

**Reasons To Reject:**

I have some concerns regarding the lack of comparison to other pathfinding algorithms, even those that baseline A* itself surpasses, and questions about whether SearchFormer can make the same guarantees that baseline A* can, and how that can be demonstrated.  I also have some questions regarding relevance to the Conference on "Language Modeling."

---

> ### Author Rebuttal · Authors · 2024-05-31
>
> We thank the reviewer for the helpful comments and feedback.
>
> **Re. guarantees for finding a shortest path:** Indeed the property that every plan is optimal is captured implicitly in the training data, and imitated by the Searchformer. Note that even trained with optimal paths, Searchformer is still not guaranteed to output optimal plans for unseen tasks. For more discussion on theoretical guarantees, see *“Re Q2 and Re Reason to Reject Pt. 2”* part in our response to reviewer KTvv (sorry, can't fit here due limited space)
>
> **Re. other planning algorithms:** Our method is a novel training paradigm, and is orthogonal to other search/planning algorithms. See *“Re Q1”* part in our response to KTvv about applying our method to different algorithms. We will incorporate this argument into Sec 3.1 and add a comment in the future work section that studying other algorithms is left for future work as it is beyond the scope of the paper.
>
> **Re. relevance to the Conference on Language Modelling:** Our motivation is that existing LLMs perform poorly on most planning tasks, and we believe that Searchformer is an important step to improve planning capabilities in the future. What we demonstrate is how to express planning (the computation involved in obtaining an optimal plan as well as the optimal plan itself) in a synthetic language. Then, we show how to train Language Models on this synthetic language and optimize the language itself by using the trained LM to reduce the number of search steps needed to arrive at an optimal plan. Therefore, we believe that our manuscript contributes to research on language models as it builds a connection between language modeling, planning, and sequential decision making. As a future work, it may be possible to apply our method to language-based search and planning traces, or perhaps to tasks where the input and output sequences are in natural language while the search trace is still in synthetic language.
>
> This argument is also highlighted in your comment about reasons to accept.
>
> We will revise the introduction and future work sections to highlight this important point.

---

> ### Author Response · Authors · 2024-06-04
>
> Thank you again for the thoughtful feedback and comments. We hope that our responses address your questions.
>
> As we are approaching the end of the discussion period, we hope you can consider raising the score to ensure the paper is accepted into the conference.

---

> ### Comment · Reviewer_6qVT · 2024-06-04
> **Thank you for the response - I will raise my score to a 7 subject to commitment to certain revisions**
>
> I would like to thank the authors for taking the time to prepare a rebuttal and to reply to the questions raised.
>
> I do not think the authors' rebuttal refutes the concerns I raised, such as lack of guarantees and lack of comparison to other methods (a concern also raised by other reviewers). However, I do see the point about LLMs and planning and the connection is solid there, in my opinion.
>
> Overall, I think the novelty of this approach and its likely interest to the community outweigh the limitations. Given the novelty, it would be understandably hard to fit every required ablation and experiment into a single paper. I am comfortable raising my score to a 7 subject to a commitment by the authors to expand the (currently very short) limitations section to include an acknowledgment of the lack of guarantees provided by Searchformer, and a mention that in this respect it is inferior to out of the box A*. Where Searchformer outperforms classical planning methods is in other areas, so this limitation is just fine given what we know about how Transformers and neural methods work generally.
>
> Please also revise the introduction and future work sections as you mentioned.
>
> If the authors would please leave a comment committing to the expansion of the limitations section along the lines outlined above, I will officially raise my score to a 7.

---

> > ### Author Response · Authors · 2024-06-04
> >
> > Thank you again for your response and assessment.
> >
> > We are committed to revising the submitted paper and expanding the limitations, future work, and introduction sections along the lines outlined above.

---

> > > ### Comment · Reviewer_6qVT · 2024-06-04
> > > **Score raised to 7**
> > >
> > > I have raised the score to a 7. Thank you for your response and your contribution.

---

### Official Review · Reviewer_Loks · 2024-05-11

**Rating:** 6
**Confidence:** 5
**Ethics Flag:** 1

**Summary:**

The manuscript proposes a framework for optimising Transformer-based models for search-based planning tasks, through iteratively fine-tuning the models on traces generated from effective (albeit heuristic) symbolic planners like A*, in order to surpass those heuristic approaches in terms of number of search steps.

**Questions To Authors:**

N/A — see above.

**Reasons To Accept:**

- The paper is well-written.

- The manuscript considers an interesting subject — improving the inductive bias of Transformer-based models. This is potentially impactful, as they remain a component of larger-capacity pre-trained models, like foundation models.

**Reasons To Reject:**

Section 1 — On the subject of potential impact: the manuscript presents a framework for improving the effectiveness of Transformer-based models and later asserts that the method can be integrated into other frameworks that similarly optimise LLMs. However, the manuscript never showed whether this is even necessary (e.g., for the tasks it considers). The manuscript should also provide comparisons with LLMs that learns A* search dynamics, in context, for reference.

Section 2 — Rather than highlighting only the works that leverage Transformers to generate single actions, the manuscript may also discuss and compare against hierarchical approaches as well as works that leverage Transformer-based models for generating plans and trajectories. Whereas the manuscript makes reference to some of these works in Section 5, it is attempting to derive its novelty by drawing a comparison between its multi-step planning capability and single action generation. Therefore, it does not seem appropriate that the manuscript should omit the other methods from this discussion.

Section 4.3 — The manuscript states, "Using this Searchformer model, we subsequently generate another short sequence dataset and repeat the fine-tuning procedure to further improve the model." How many times, and — more importantly — why? What general insight can the manuscript provide about, e.g., the relationship between the nature of the task and the number of fine-tuning steps that would be required to obtain the best Searchformer model?

---

> ### Author Rebuttal · Authors · 2024-05-31
>
> **Re Sec. 1:** We aim to improve the performance of Transformers in complicated reasoning/planning tasks, which serve as LLM's core components. We train from scratch to make clean-cut comparison, since many pre-trained models are trained on proprietary datasets to excel in major benchmarks, which may contain search dynamics of puzzles. We do not assert that Searchformer can be integrated into other frameworks, but suggest the possibility (Sec. 5.2 future work).  Our initial tests show that in-context learning with SoTA LLM cannot solve our tasks (will put in Appendix).
>
> **Re Sec. 2:** Our approach is different from hierarchical planning, in which it typically defines temporal abstractions (eg., options) and relies on end2end training to learn such abstraction from scratch via RL. In contrast, our approach starts with imitation learning on well-defined steps of A*’s search process, and improve via fine-tuning. Therefore, unlike hierarchical planning that may be very tricky to tune and learn, and may require a lot of data, our approach may be more data efficient. We will include the discussion in the revision.
>
> **Re Sec. 4.3:** Our bootstrapping method finds a shorter execution trace by repeatedly sampling sequences with the same prompt. These shorter execution traces will still lie in the support of the distribution modeled by the trained Searchformer. By repeating the bootstrapping process, we aim to shift the execution trace length distribution to discover new execution traces that lie outside the initial A* training dataset. This shift can be observed in Fig 10a, where most points lie below the main diagonal indicating that the execution trace length distribution has been shifted significantly after multiple iterations.
>
> At this point, it is not clear how many bootstrapping iterations are optimal. Variance in execution trace length of the initial training data may play an important role. For example, using an inefficient search heuristic will lead A* to explore larger portions of the search tree until an optimal solution is encountered by random chance, leading to a higher variance in the resulting trace dataset. In this case, improving the Searchformer model should be easier. If the training data already outlines an optimal search process, it may be much more difficult for an improvement.
>
> One iteration currently takes more than 24 hours to complete while using 512 V100 GPUs. So we leave a more detailed study of additional iterations for future work.

---

> > ### Comment · Reviewer_Loks · 2024-06-05
> > **Official Comment by Reviewer Loks**
> >
> > I appreciate the authors' responses; however, I do not feel as if the response alleviated all of my concerns.
> >
> > It remains difficult to assess the efficacy of the approach, relative to alternatives, since direct comparisons with strongly motivated LLM-based and/or symbolic planning approaches are missing.
> >
> > Without an analysis on optimality/completeness (requested by Reviewer KTvv) or on the relationship between some characterization of the task and the number of fine-tuning steps, it remains difficult to extract useful insights from the proposed approach that can guide an investigation in a different problem or domain.
> >
> > I will retain my current score.

---

> ### Author Response · Authors · 2024-06-04
>
> Thank you again for the thoughtful feedback and comments. We hope that our responses address your questions.
>
> As we are approaching the end of the discussion period, we hope you can consider raising the score to ensure the paper is accepted into the conference.

---

### Official Review · Reviewer_KTvv · 2024-05-25

**Rating:** 6
**Confidence:** 3
**Ethics Flag:** 1

**Summary:**

This paper presents a novel and effective approach to train Transformer models for complex planning tasks and demonstrates improvements over traditional algorithms like A*. The search dynamics bootstrapping methodology, which trains the model to imitate A*'s
search process and then fine-tunes it to discover shorter solution sequences, is a creative and promising technique. And it could be extended to other algorithms and problem domains, paving the way for more efficient and generalizable neural network-based planners.

**Questions To Authors:**

1. Could this method be applied to more advanced symbolic planners or other
types of heuristic search algorithms?
2. Is there a theoretical bound on the efficiency gains that can be achieved
through this bootstrapping process?

**Reasons To Accept:**

Pros:
1. The paper presents a new method, search dynamics bootstrapping, for
training Transformer models to solve complex planning tasks, which opens
up new possibilities for applying deep learning to planning and reasoning
problems.
2. The Searchformer model can outperform A* algorithm in efficiency and
problem-solving ability, which highlights the potential of this approach to
planning processes and tackling more complex problems.
3. This paper presents a well-designed set of experiments that systematically
test their approach under different conditions. The use of synthetic
datasets and the Sokoban domain demonstrate the approach's
effectiveness, while also suggesting its potential for generalization to other
planning problems.

**Reasons To Reject:**

Cons:
1. While the paper demonstrates improvements over the A* algorithm, it does
not compare the Searchformer to other recent advances in neural network-based
planning or symbolic models, making it difficult to assess its relative
performance.
2. The paper does not provide theoretical guarantees on the optimality or
completeness of the Searchformer's planning process. Further analysis is
needed to understand the conditions under which the model's performance
can be guaranteed.
3. The search dynamics bootstrapping approach relies on the A* algorithm to
generate the initial training data. If the underlying algorithm is suboptimal or
fails to find solutions for certain problems, it could limit the performance of
the Searchformer model.

---

> ### Author Rebuttal · Authors · 2024-05-31
>
> **Re Q1:**
> Our method could be applied to different planning algorithms by designing a new synthetic language. For A*, we found it is sufficient to express all of its steps in a token sequence. Given any prefix of the token sequence, the state of A* can be reconstructed to resume execution. This recipe of logging all operations could be applied to different algorithms to construct the corresponding synthetic language and train Searchformer.
>
> **Re Q2 and Re Reason to Reject Pt. 2:**
> A limitation of most deep learning methods is that guaranteeing test performance is often very difficult, and the theoretical properties of transformers still remain wide open. This applies to both (1) theoretical guarantees of planning like what A* gives, and (2) whether the bootstrapping process has guarantees. The motivation of this work is not for theoretical guarantees, but to open a new door to discover more efficient algorithms (in terms of “number planning of steps”, not wall-clock time) than traditional ones, verified by extensive empirical analysis.
>
> For bootstrapping, we choose to use non-deterministic sequences to diversify the generations, which makes it easier to pick up better planning samples for the next iteration. The fine-tuning only needs a reward function (shortening the search trace while maintaining optimality), and does not require hand-crafted heuristics, as in traditional methods.
>
> **Re Reason to Reject Pt. 1:** Sec 2 provides extensive literature review. In summary, our work provides a novel training paradigm to improve traditional search methods by first doing imitation learning on search paths, and then further improving them by leveraging differentiability of Transformers. In comparison, previous works either start from sequence models (e.g., pre-trained LLMs) and explore heuristics for better reasoning in the natural language space (e.g., Chain-Of-Thoughts), or learn heuristics for existing search procedures (e.g., AlphaGo).
>
> **Re Reason to Reject Pt. 3:** Our approach should still work in this case with minor changes. If the training data have suboptimal solutions, we can generate multiple sequences per input, and pick the one that is closer to the optimal plan in length (and have shorter search trace), and use that data for the next iteration. This will lead to better plans that is close to optimal. In the paper, we focused only on the search trace length as we are interested in optimizing the internal search and planning process of the model.

---

> ### Author Response · Authors · 2024-06-04
>
> Thank you for the thoughtful review and comments. We hope that our responses address your questions and we will revise the submitted paper in the corresponding sections to integrate the discussed points.
>
> As we are approaching the end of the discussion period, we hope you can consider raising the score to ensure the paper is accepted into the conference.

---

### Decision · Program_Chairs · 2024-07-10

**Decision:**

Accept

**Comment:**

The manuscript presents a recipe for training Transformers to perform planning and shows that the resulting models can effectively perform planning for task requiring complex action sequences. Fine-tuned, the model can sometimes outperform A*, the oracle algorithm that was used to seed training.
The manuscript offers a new take on how to use Transformer models in service of planning algorithms. Traditional approaches such as MCTS and A* use handcrafted rules for building and refining search trees. Using network models to learn such rules is an interesting and underdeveloped direction of research. While search is initially learned from A* data, they develop a mechanism for bootstrapping search dynamics that creates better planners that take fewer steps to reach solution on Sokoban. The paper is written clearly with well executed analysis and figures.The manuscript could be even more compelling if comparisons to other network-based planning approaches.More difficult benchmarks would always be a plus, but the paper develops an interesting idea clearly and Sokoban is a difficult RL/control problem on its own even though it is situated in a gridworld environment.
On the whole, the simple and compelling results make the paper easy to read and will inspire more research into the underexplored direction of using large networks to perform planning algorithms.